# Pick’s Tau Fibril Shows Multiple Distinct PET Probe Binding Sites: Insights from Computational Modelling

**DOI:** 10.3390/ijms22010349

**Published:** 2020-12-31

**Authors:** Sushil K. Mishra, Yoshiki Yamaguchi, Makoto Higuchi, Naruhiko Sahara

**Affiliations:** 1Advance Glycoscience Research Cluster, National University of Ireland Galway, H91 W2TY Galway, Ireland; skmishra@olemiss.edu; 2Faculty of Pharmaceutical Sciences, Tohoku Medical and Pharmaceutical University, Sendai, Miyagi 981-8558, Japan; 3National Institute of Radiological Sciences, National Institutes for Quantum and Radiological Science and Technology, Chiba 263-8555, Japan; higuchi.makoto@qst.go.jp

**Keywords:** docking, MM/GBSA calculation, positron emission tomography (PET) imaging, tau, PET tracers

## Abstract

In recent years, it has been realized that the tau protein is a key player in multiple neurodegenerative diseases. Positron emission tomography (PET) radiotracers that bind to tau filaments in Alzheimer’s disease (AD) are in common use, but PET tracers binding to tau filaments of rarer, age-related dementias, such as Pick’s disease, have not been widely explored. To design disease-specific and tau-selective PET tracers, it is important to determine where and how PET tracers bind to tau filaments. In this paper, we present the first molecular modelling study on PET probe binding to the structured core of tau filaments from a patient with Pick’s disease (Tau^PiD^). We have used docking, molecular dynamics simulations, binding-affinity and tunnel calculations to explore Tau^PiD^ binding sites, binding modes, and binding energies of PET probes (AV-1451, MK-6240, PBB3, PM-PBB3, THK-5351 and PiB) with Tau^PiD^. The probes bind to Tau^PiD^ at multiple surface binding sites as well as in a cavity binding site. The probes show unique surface binding patterns, and, out of them all, PM-PBB3 proves to bind the strongest. The findings suggest that our computational workflow of structural and dynamic details of the tau filaments has potential for the rational design of Tau^PiD^ specific PET tracers.

## 1. Introduction

Filamentous tau aggregates are the histopathological hallmark of a group of neurodegenerative diseases including Alzheimer’s disease (AD), progressive supranuclear palsy (PSP), corticobasal degeneration (CBD), Pick’s disease (PiD), and familial frontotemporal lobar degeneration with underlying tau pathology (FTLD-Tau) collectively referred as tauopathies [1]. The conformational diversity of tau assemblies derived from distinct isoform compositions, posttranslational modifications and interactions with other molecules results in variations in pathological tau depositions in different subcellular compartments, cell types, and brain regions [2]. The composition of tau fibrils in each of the tauopathies is well characterized biochemically and by pathological examination. Paired helical filaments (PHFs) in AD are composed of all six tau isoforms, whereas straight filaments (SFs) and twisted ribbons are made up of isoforms with four repeat (4R) domains in PSP and CBD, and with three repeat (3R) domains in PiD [3]. Electron cryo-microscopy (cryo-EM) has revealed high-resolution structures of the C-terminal core region in tau filaments from AD, PiD, CBD and chronic traumatic encephalopathy (CTE) brains [4,5,6]. Analysis of samples, from multiple cases of sporadic and inherited AD, has revealed a common structure of the filament cores [7]. The Tau^AD^ filaments are composed of two C-shaped protofilaments arranged either symmetrically base-to-base (PHF) or back-to-base (SF). The cores in both types of filament consist of 73 residues (V306–F378) with eight β-strands (β1–β8). On the other hand, distinct tau filament structures are found in PiD, CBD or CTE subjects [5,6,8]. Tau^PiD^ core is formed of 94 residues (K254 to F378), and is, therefore, longer than Tau^AD^. A total of nine β-strands (β1–β9) is arranged in four cross-β packing stacks in Tau^PiD^. These cryo-EM studies raise the interesting question as to whether different tau filament structures can be distinguished by tau tracers, which have recently been developed for in vivo positron emission tomography (PET) imaging.

Tau tracers are mostly designed according to β-sheet binding properties and theoretically label all filamentous tau aggregates (e.g., neurofibrillary tangles (NFTs), neuropil threads, tufted astrocytes, astrocytic plaques, and coiled bodies). Several PET tracers for detecting filamentous tau inclusions, such as [^11^C]PBB3, [^18^F]PM-PBB3, [^18^F]AV1451, [^18^F]THK5351, [^18^F]MK-6240, [^18^F]R06958948, [^18^F]GTP-1, and [^18^F]PI-2620, were developed and have been applied to human subjects [9,10,11,12,13,14,15,16]. The distribution of the bound tracers recapitulated Braak NFT staging [17] in AD brains, suggesting in vivo imaging-based staging of tau pathologies. On the other hand, [^18^F]AV1451 and [^18^F]THK5351 exhibit off-target effects such as binding to monoamine oxidase (MAO)-A and MAO-B, respectively [18,19]. In particular, the binding of [^18^F]THK5351 to MAO-B is strongly suggested in the detection of MAO-B-positive astrogliosis [20,21,22]. Second generation tracers ([^18^F]PM-PBB3, [^18^F]MK-6240, [^18^F]R06958948, [^18^F]GTP-1, and [^18^F]PI-2620) seem to have less off-target binding [12,23,24]. More recently, the binding properties of these PET tracers with PHFs and SFs were examined by computational modeling using structural information from cryo-EM studies [25]. Tau^PiD^ folds in a J-shaped structure, instead of the C shape seen in patients with AD [4,8]. The modeling identified three to four potential high-affinity binding sites for these tracers with some preference for certain sites over others for each tracer [25,26]. Importantly, validation for the off-target interaction of tau tracers with amyloid β fibrils was also undertaken [25]. However, modeling of PET tracer binding sites in non-AD tauopathies has not been explored.

Of the various neurodegenerative diseases, Pick’s disease is a rare type of age-related dementia that affects the frontal lobes of the brain, resulting in speech problems like aphasia, behavior difficulties and eventually death. Pick’s tau folds quite differently from Alzheimer’s tau filaments [4,8], and its surface residues where the tracers potentially bind are very different too. Since the binding specificity of current tau PET probes have not been completely validated against tau pathologies, approaches employing the structure-guided design of tracers for different filaments can be extremely useful. In order to elucidate the binding mechanism of PET probes in Tau^PiD^, we performed computational modeling to characterize the binding of various PET probes to Tau^PiD^, and obtained insights into their unique binding properties.

## 2. Results and Discussion

### 2.1. Assessing Binding Regions and Affinity of PET Tracers to Tau^PiD^ by Docking

Current tau PET tracers such as [^18^F]AV-1451 ([^18^F]flortaucipir), [^18^F]THK5351, [^11^C]PBB3, [^18^F]PM-PBB3 and [^18^F]MK-6240 (Figure 1A) have been successful in the in vivo evaluation of tau pathology of AD brains [9,11,16,27,28]. [^18^F]PM-PBB3 PET imaging also was demonstrated to be useful for the differential diagnosis of non-AD dementia patients (e.g., PSP, CBD, and Pick disease) [16]. However, binding of other tracers to tau filaments derived from non-AD brains is still under investigation. For evaluation of each of the tracers, we first tested our docking strategy on Tau^PiD^ using the structural information on Tau^PiD^ from the Protein Data Bank (PDB ID: 6GX5) [8]. PET tracers are designed to bind to cross-β structure [29]; therefore, we assumed that PET tracers bind to the C-terminal core region of tau filaments and not to the other flexible region [4,8]. The docking results show that a number of surface binding sites (S1–S11 in Figure 1B) in Tau^PiD^ can potentially bind PET probes. AutoDock Vina docking scores (hereafter called docking scores) and X-Score rescoring energies (hereafter called rescoring energies) of the highest-scoring pose in each binding site are listed in Table 1. The inner cavity formed between P312-V313-D314 and P332-G333-G334 (C1 in Figure 1B) showed the strongest affinity for all PET probes.

Below, we primarily focus on the solvent-accessible surface binding of PET probes and then examine the cavity binding using a bigger tau assembly. In the case of the surface binding sites, docking scores and rescoring energies of the same binding pose show a similar trend for all the PET tracers.

The predicted docking scores of the PET probes at surface sites are in the micromolar range (binding free energy of −8.2 kcal·mol^−1^ corresponds to *K_d_* of 10 μM), and are in agreement with other docking studies [25]. Docking scores for MK-6240 (−7.4 kcal·mol^−1^ in S6), PM-PBB3 (−6.1 kcal·mol^−1^ in S8), and PiB (−7.1 kcal·mol^−1^ in S7) at surface binding sites are weaker compared to AV-1451 (−9.4 kcal·mol^−1^ in S7), PBB3 (−8.4 kcal·mol^−1^ in S7) and THK-5351 (−8.3 kcal·mol^−1^ in S7) (Table 1). Although PiB is used for imaging *β*-amyloid (Aβ) plaques, this compound has a docking score similar to that of the Tau PET probes (MK-6240 and PM-PBB3).

Rescoring energies predict PBB3 to be the most potent binder at a surface binding site (S7; −8.7 kcal·mol^−1^). Otherwise, rescoring energies for the surface binding sites trend like docking scores, in that AV-1451 (−8.4 kcal·mol^−1^ in S7) and THK-5351 (−8.3 kcal·mol^−1^ in S7) bind with similar strength and somewhat more strongly than PM-PBB3 (−7.5 kcal·mol^−1^ in S8), MK-6240 (−7.7 kcal·mol^−1^ in S6) and PiB (−7.8 kcal·mol^−1^ in S7) (Table 1). Among all surface sites, PM-PBB3, THK-5351 and PiB have the weakest rescoring energies, which is almost in line with the docking scores (Table 1). Four PET probes, not MK-6240 and PM-PBB3, prefer surface binding site S7. PM-PBB3 and MK-6240 bind neighboring S8 and S6, respectively, with the highest affinity. Only PBB3 and PiB bind to S1 and S3, but with weak affinity. S11 uniquely binds only PM-PBB3 and PiB.

Docking scores and rescoring energies are in good agreement when the scores of individual PET probes are compared in multiple binding sites. However, there are some inconsistencies when comparing the scores of different PET probes with each other. Since docking is performed to a rigid protein, it fails to take into account protein flexibility and solvent effects, and the resulting docking poses can be artificial [30]. Therefore, we further validated the predicted docking poses by MD simulations and binding free-energy calculations.

### 2.2. Molecular Dynamics and MM/GBSA Calculations

We performed MD simulations of each PET probe bound to Tau^PiD^ simultaneously in a single simulation over a range of high to low affinity binding sites as predicted by docking. Since in each individual MD simulation, the same PET probe is present in all predicted binding sites, neighboring PET probes can influence binding, as expected in reality. In the Tau^PiD^/PET probe complexes, compounds present at surface binding sites sometimes drifted out from their binding site to bulk solvent and then started interacting with another probe that is already bound to tau (Appendix A). Therefore, for the Molecular Mechanics/Generalized Born Surface Area (MM/GBSA) binding energy calculation in each site, PET probes in all the remaining binding sites were stripped off by pre-processing the MD trajectory, otherwise the energy contribution arising from probe-probe interaction could be misleading. Thus, there is no influence of neighboring probes in the MM/GBSA binding-energy calculation and the presented values solely reflect intermolecular interactions between PET probe and tau at that particular binding site.

Tau^PiD^ was quite stable during the MD simulations, except that the fibrils at both edges start to unfold during at extended times. Unfolding could be due to interaction of the fibril edges with the solvent. In a proper tau assembly, filaments would not be exposed to solvent; instead, they are stacked between other fibrils. Since Tau^PiD^ fibrils exist in different polymorphic forms, fibrils are expected to be highly flexible. The MD-based MM/GBSA approach used here for binding free-energy calculations takes into account tracer and receptor flexibility issues and, therefore, is expected to provide more realistic binding energies than docking or rescoring functions. We refer to binding enthalpies as binding energies here, because we assume that entropic contributions do not play a major role in predicting relative binding order for the same system. The computed binding energies of the PET probes in each binding site are shown in Figure 2 and listed in Table 2.

MM/GBSA binding calculations on 1000 frames suggest that Pick’s tau core has multiple surface binding sites (see Figure 1B) where S7 is the most potent for AV-1451 (−25.5 kcal·mol^−1^), THK-5351 (−22.8 kcal·mol^−1^) and PiB (−25.9 kcal·mol^−1^), and S8 for PBB3 (−27.9 kcal·mol^−1^) and PM-PBB3 (−34.2 kcal·mol^−1^) (Table 2). PBB3, which has been used to visualize tau deposition in AD and PiD, shows binding to multiple Tau^PiD^ surface sites with almost equal strength (−26.0 kcal·mol^−1^ in S1; −26.2 kcal·mol^−1^ in S5; −24.3 kcal·mol^−1^ in S6; −27.6 kcal·mol^−1^ in S7 and −27.9 kcal·mol^−1^ in S8) and could be a potent Tau^PiD^ tracer (Figure 2). PM-PBB3 also binds in a site (S2n) neighboring site S2 with binding energy −30.2 ± 3.2 kcal·mol^−1^ (Table 2). PM-PBB3 also docked to site S3n (neighbor of S3) but binding energies could not be extracted due to van der Waals clashes in some of the snapshots. Site S8 is unique for both PBB3 and PM-PBB3 as only these show binding here (all other PET tracers either failed to bind here or they drifted away from the binding site during the MD simulation) (Table 2, Figure 2).

MK-6240 showed very weak affinity for surface sites, which is due to the fact that the probe drifted away from all the predicted surface sites during the simulation (Figure 2). Unlike other PET probes, the structure of MK-6240 is not planar in 3D and cannot bind to narrow hydrophobic patches formed between two amino acid side chains. MK-6240 seems an unlikely candidate as a Tau^PiD^ tracer. PBB3 and PM-PBB3 bind most strongly to Tau^PiD^, either at S7 or neighboring sites. AV-1451, THK-5351 and PiB effectively bind to S7 only; other binding sites are too weak to be significant. It has been reported that PBB3 strongly binds to Pick bodies in brain slices while AV1451 only weakly binds [31]. Thus, S8 might be the high-affinity site for PBB3 and PM-PBB3 under physiological condition, which needs to be verified in future structural study.

MM/GBSA calculations have the capability to provide per-residue decomposition of the energy contribution to the binding free-energy, and this can provide insight into the binding of the PET tracers to tau, namely that it is primarily derived from CH/π stacking interactions (Appendix A). For example, in sites S7 and S8, PET probes bind to hydrophobic patches formed by sidechains of K347 and R349, respectively, via stacking interactions between hydrogen atoms of the amino-acid side chain and π electrons in the PET tracers (Figure 3). The CH/π stacking interactions are a strong attractive molecular force occurring between a soft acid and a soft base and are frequently seen in protein–glycan complexes [32,33,34].

In general, the binding pattern shown by the MM/GBSA energies is quite similar to the docking scores in most cases, but there are significant differences in some. Surface binding of PET probes at S7 is predicted to be the strongest by both docking and MM/GBSA calculations. On the other hand, PBB3 binding to S1 is predicted to be very weak by docking but quite strong by MM/GBSA energies. Two factors likely cause these discrepancies. Firstly, due to the narrow range of binding scores of AutoDock Vina, computational differentiation between weak and strong binding sites is constrained. Thus, a small error in docking scores can affect the rankings. Secondly, the differences in MM/GBSA energies are within the range of standard deviation of the energies over the 1000 frames used in the calculations. In this context, MM/GBSA energies are a more useful parameter as they also provide a meaningful statistic that can be taken into account to rank the binding strength of PET tracers in multiple binding sites.

### 2.3. Comparison of Probe Binding Affinities between Tau^AD^ and Tau^PiD^

Figure 4A shows PET probe binding sites in Tau^AD^ first reported by Goedert et al., [26] and soon after by Murugan et al. [25], as well as the sites on Tau^PiD^ from this study. From a structural stance, it can be seen that the binding sites of the majority of the PET probes in Tau^AD^ and Tau^PiD^ are formed in cavities surrounded by a basic residue (Lys, Arg) either from one or from both sides (Figure 4A,B). Docking and MM/GBSA energies of PBB3, THK5351 and MK-6240 with Tau^AD^ were published by Murugan et al. [25] and are here compared with Tau^PiD^. Docking scores and MM/GBSA approach-based binding energies of PBB3, THK5351 and MK-6240 at their strongest surface binding site within Tau^AD^ and Tau^PiD^ are shown in Figure 4C–F. The figures suggest that these PET probes have a greater binding preference for the Tau^AD^ fibril than that of Tau^PiD^ (Figure 4E,F). The study by Murugan et al. [25] lacks the name of the docking software and details of the Generalized Born (GB) model used in the MM/GBSA calculation. Since different models in AMBER18 lead to a significantly different range of absolute binding energies, the deviations between binding energies for two different types of tau filaments could be due to a difference in the length of the MD simulation (100 ns) or to a different GB model chosen for the MM/GBSA calculations. We chose longer MD simulations (100 ns) to allow sufficient time for dissociation. Despite expected differences in absolute energies, the relative binding order of three common PET tracers is the same in both studies (PBB3 > THK5351 > MK-6240) (Figure 4C,D). Binding energies from the MM/GBSA calculation also show the same binding preference (PBB3 > THK5351 > MK-6240) in Tau^AD^ and Tau^PiD^ (Figure 4E,F), suggesting that a discussion of relative binding order is still valid despite a possible different choice of MM/GBSA parameters.

### 2.4. PM-PBB3 as a Potent PET Tracer for Detecting Tau^PiD^

Most recent findings with both clinical and pre-clinical studies of [^18^F]PM-PBB3 imaging revealed the feasibility of using PM-PBB3 to recognize tau pathologies in both AD and non-AD brains including Pick’s disease [16]. To confirm PM-PBB3 binding to Tau^AD^ and Tau^PiD^ and to determine preference through binding affinity, we calculated binding affinities of PM-PBB3 for both Tau^AD^ and Tau^PiD^ by docking and MM/GBSA calculations. Absolute MM/GBSA energies of PM-PBB3 are similar for both Tau^AD^ and Tau^PiD^ (−34.2 kcal·mol^−1^ for Tau^AD^ and −34.3 kcal·mol^−1^ for Tau^PiD^) (Figure 5A,B), confirming that PM-PBB3 binds to both kinds of tau filaments. This is in the agreement with recent experimental study that indicates binding of PM-PBB3 to Tau^PiD^ and Tau^AD^ is nearly similar [16]. On the other hand, the relative binding order of PM-PBB3 illustrates unique features of the Tau^PiD^ protofibril when compared to that of Tau^AD^. Among PET tracers used in this study, the binding energy of PM-PBB3 relative to PBB3, THK-5351, MK-6240 and other PETs is relatively strong for Tau^AD^ compared to Tau^PiD^. Since docking cannot predict experimental binding trends of PM-PBB3 so accurately, MD based free-energy approaches are important to consider in the rational design of probes for disease specific tau filaments. In addition, this approach can be applied to the interaction of PET tracers with non-tauopathy off-targets, such as MAO-A/-B and other dementia/neurodegenerative proteins (amyloid-β or α-synuclein). Since previous studies confirmed relatively low affinities of tau tracers, [^11^C]PBB3 and [^18^F]PM-PBB3, for MAO-A/B and α-synuclein by in vitro radioligand binding assay [16,35,36], it will be interesting to perform side-by-side validation with both an in vitro assay and an MM/GBSA approach.

Our simulation studies demonstrate that PM-PBB3 has the highest MM/GBSA binding energy of the seven compounds tested, and revealed that the surface binding sites in Tau^PiD^ are rather unique. S8 site is highly selective for PM-PBB3 over PBB3 (Figure 2). In addition to these predicted features, PM-PBB3 has several practical advantages, e.g., greater metabolic stability compared with PBB3 [16]. Furthermore, a ^18^F-labeled tracer has the advantage of broader availability and higher throughput than ^11^C-labeled tracers. Taking all these points together, PM-PBB3 stands out as the most promising tracer for diagnostic imaging of diverse neurodegenerative tauopathies.

### 2.5. Does Pick’s Tau have a Potential Cavity Binding Site?

Unlike Tau^AD^, Tau^PiD^ shows a potential cavity binding-site (C1) where PET probes bind within a cavity formed between P312-V313-D314 and P332-G333-G334 (Figure 1B). The PET tracers AV-1451 (−11.0 kcal·mol^−1^), MK-6240 (−11.0 kcal·mol^−1^), PBB3 (−10.1 kcal·mol^−1^), PM-PBB3 (−10.0 kcal·mol^−1^), and THK-5351 (−10.3 kcal·mol^−1^) have comparable docking scores in the C1 site, but PiB (−8.9 kcal·mol^−1^) binds comparatively weakly. Of all the docking scores, those of PiB in C1 and at the surface sites are weakest. AV-1451, PBB3, and THK-5351 bind with similar strength to C1, but MK-6240 and PiB show weak affinity. Binding energies from MD simulations followed by MM/GBSA calculations on 1000 frames suggest that binding site C1, which is formed due to folding of the fibril, is a favored binding site. All seven compounds show stable binding to C1, and the binding is stronger than it is to surface binding sites S1 to S11 (Figure 6). This observation is similar to PET probe binding to β-amyloid, where cavity sites also bind to PET probers with higher affinity than surface sites [25]. Of the tracers, PBB3 binds strongest to C1—an interesting observation as the binding is stronger than PiB. The C1 site primarily consists of hydrophobic interactions mediated by the hydrogen atoms of P312 and P332.

To further explore the cavity site C1 unique to Tau^PiD^, we performed 1-μs MD simulation of a bigger tau assembly composed of a 49 polypeptide (Tau49^PiD^) without any PET probes in the binding sites. We further performed a Tunnel analysis around C1, in a 1-μs MD trajectory to see if the tunnel is realistic and maintained during the MD simulation. The tunnel profile in the cryo-EM structure of Tau^PiD^ shows two potential tunnels with some slightly narrow entry paths between filaments to C1 (Figure 7A and Appendix A). We further evaluated protein tunnels on 500 conformations of a bigger Tau^PiD^ assembly (Tau49^PiD^) taken at every 2 ns from 1-μs long MD simulation (Figure 7A–F). Tunnel lengths and radii in the MD-simulated structure vary during the simulation, but are very narrow compared to the cryo-EM structure. A cluster analysis of all the tunnels in those 500 snapshots shows clusters mainly in two distinct regions in the plot (Figure 7G). The majority of tunnels cluster in a region varying in length from 150 to 300 Å and number 60 to 150. The distance between P213 residues at both ends of the Tau49^PiD^ is ~230 Å, indicating that some of the tunnels are >230 Å long. There may be interlinking tunnels (Appendix A). The other cluster shows smaller tunnels of less than 50 Å in length but higher in number. Therefore, the majority of conformers possess a tunnel of about 150 to 250 Å in length. The radius profile suggests that tunnels shrink during MD simulation (Appendix A). However, a number of snapshots still show continuous tunnels connecting both ends of the assembly, often merging to one or exiting to the surface.

It is essential to mention here that tunnel finding algorithms have limitations and it is expected that a program will identify an exit towards the surface where the sphere radius for moving forward in the tunnel is lower than the radius to exit towards the surface. Some of the MD frames show tunnels starting at one end of the assembly and finishing at the other end (Figure 7E). However, very often a tunnel either ends inside the assembly or finds an alternative path through a neighboring tunnel (Figure 7F) and then connects back to the original tunnel in the C1 region.

Our overall analysis of tunnels in the MD simulation snapshots suggests that there may be a potential cavity binding site in Tau^PiD^. However, very narrow circumferences in several snapshots and reductions in tunnel radii in the large tau assembly also suggest that the broad cavernous region around C1 could be a result of limitations in the cryo-EM structure. The cryo-EM structure of Tau^PiD^ was solved with a resolution of 3.2 Å and it has a distance of 8.5 Å between the P312 and P332 sidechains (Figure 7H) where C1 is formed. Ideally, the cavity binding site needs to be verified experimentally. Recent cryo-EM studies of CBD and CTE tau filaments indicate the presence of additional non-proteinous densities inside the cavity [5,6]. Our computational approach will also contribute to the understanding of such internal binding of non-proteinous molecules once the structure is identified.

In summary, the computational workflow presented in this study can be a basis for the rational design of tracers specific to a particular tau fiber. Currently, no atomic data are available to experimentally show the binding mode between PET tracer and tau fibril. Docking studies are complementary to 3D structural analyses, such as cryo-EM, which usually provide averaged electron densities at low-to-medium resolutions. The strong point of the MM/GBSA approach is the ability to estimate individual binding energies (binding constants) of multiple binding sites such as those that exist in the tau fiber. Experimental approaches cannot provide binding constants for each of such multiple binding sites. Although a gap still exists between theoretical predictions and experimental results, a computational approach will ensure advances in elucidating tracer-protein interactions of multiple binding sites.

## 3. Materials and Methods

### 3.1. Structure Preparation

The structures of Tau^PiD^ (PDB ID: 6GX5) and Tau^AD^ (PDB ID: 5O3L) were taken from the Protein Data Bank [8]. Tau^PiD^ and Tau^AD^ structures comprise 94 and 73 amino acids, respectively, from the total 441 residues in the 2N4R tau sequence. The reported Tau^PiD^ assembly has three tau fibrils (K254–F378), so we prepared a five-fibril assembly by superimposing structures over each other to have a structure comparable to the one used in the Tau^AD^ studies [25]. Five model compounds representing PET tracers, AV-1451, MK-6240, PBB3, PM-PBB3, and THK-5351, and the Pittsburgh compound B (PiB), were prepared using Maestro [37]. PET structures were optimized geometrically initially using the quantum mechanics (QM) approach (HF/6-31+G(d,p) level of theory and then further at B3LYP/6-31G*) in Gaussian09 [38]. The Restrained Electrostatic Potential (RESP) atomic partial charges were calculated using program *antechamber* of Amber18 [39] with ESP potentials from Gaussian.

### 3.2. Docking

The tau assembly was centered in a conformational search space of 30 Å × 30 Å × 70 Å. We chose this size to ensure that it was large enough to capture all the surface sites in the fibrils. A total of 100 docking conformations of each PET probe was obtained by software AutoDock Vina-carb using *chi_coeff* = 0 and *chi_cutoff* = 12. The used values of *chi_coeff* of and *chi_cutoff* switch Vina-carb to AutoDock Vina [40] are denoted as Vina throughout the manuscript. An exhaustiveness of 20 and energy range cutoff of 8 kcal·mol^−1^ were used. All the binding poses were refined by a rescoring function of X-Score [41]. The docking population of PET probes was calculated for each site, and all top scoring docking poses from each cluster were further analyzed by extensive MD simulations and binding free energy calculations. Possible binding sites were labelled as cavity site 1 (C1), and surface sites 1 to 11 (S1 to S11).

### 3.3. MD Simulation

A Tau^PiD^ assembly with bound PET probe in all potential binding sites was prepared. For each PET probe, we performed MD simulation of fully solvated assemblies where the PET probe is simultaneously bound to Tau^PiD^ in all the binding sites predicted by docking. For each binding site, the highest-scoring binding pose from each cluster was selected for MD simulation. The protein was treated with Amber ff14SB force field [42], whereas PET probes were modelled using General Amber Force Field v.2.0 (GAFF2) [43] with RESP charges. All the complexes were solvated by an octahedral TIP3P water box extending 12 Å from each edge of the protein. A total of 45 Cl^−^ ions was added to neutralize the whole system. A multi-step protocol published elsewhere [44] was used to equilibrate the complexes. Finally, a 100 nanosecond MD of each complex was performed at NPT, using MD settings: temperature at 300 K, temperature scaling by Langevin dynamics (collision frequency = 2), pressure relaxation every 1.2 picosecond, SHAKE constraints, nonbonded interaction cutoff of 10 Å, and 2 fs integration time step. Similarly, MD of Tau^AD^ complexed with PM-PBB3 in five binding sites was performed.

The Tau^PiD^ cryo-EM structure possesses a core cavity at a site between β-sheets containing residues P312 and G332, which can accommodate PET tracers. To check the validity of the existence of the core cavity (whether it is real or an artefact resulting from the cryo-electron microscopy approach), a 230 Å long 49 fibril Tau^PiD^ assembly (Tau49^PiD^) was prepared for MD simulation. The MD of Tau49^PiD^ in apo form was performed on the microsecond timescale. In contrast to the MD of five fibril Tau^PiD^, this Tau49^PiD^ assembly was solvated with a rectangular water box, extending 10 Å from the protein surface, and comprised a total of ~80,000 water molecules and 441 Cl^−^ ions. Prepared systems were quite large (320,000 and 400,000 atoms for the apo and PET probe-bound Tau^PiD^ structures, respectively). MD of Tau49^PiD^ fibrils was performed for 1 µs in explicit solvent using cuda version of *pmemd* in Amber18 [39]. The non-bonded interaction cut-off was set to 9 Å for these systems. The other MD settings were as described before. We analyzed the MD trajectories using *cpptraj* [45].

### 3.4. MM/GBSA Calculations

To calculate binding free energy of the PET probe in each site, solvent molecules and tracers present in remaining binding sites were removed from the trajectory. A total of 1000 snapshots were extracted after each 10 ps of a 100-ns MD simulation and used for the PET probe binding energy calculation with the MM/GBSA approach [46]. A similar procedure was repeated for the PET probe in other binding sites one by one. We equate binding enthalpies as binding energies assuming that entropic contributions do not play a key role in predicting relative order of binding for the same system [25,47]. The GB^OBC^ generalized-born model [48] was used with set mbondi2 radii. Salt concentration was 0.15 M. The surface tension and non-polar solvation free energy correction terms were set to 0.005 kcal·mol^−1^ and 0.0, respectively, for the solvent accessible surface area (SASA) calculation. An offset value of 0.0 was used to correct non-polar solvation free energy contributions. The interior and exterior dielectric constants were set to 1 and 80, respectively. Other parameters were their default values in Amber18.

### 3.5. Tunnel Calculations

Tunnels around the core-site in Tau^PiD^ were calculated using the CAVER Analyst 2.0 program [49] with a probe radius of 0.9 Å and shell radius of 3.0 Å, shell depth of 4.0 Å and a clustering threshold of 3.5 Å. Tunnels were calculated on 500 frames taken after every 2 ns from the 1-µs simulation.

## Figures and Tables

**Figure 1 ijms-22-00349-f001:**
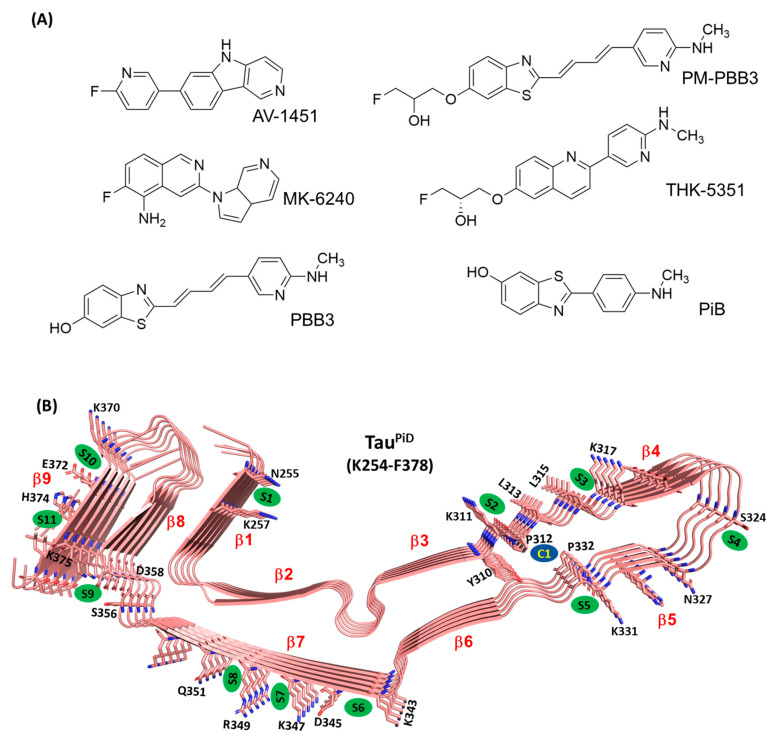
Positron emission tomography (PET) probes and their binding sites in tau predicted by molecular docking. (**A**) Chemical structures of unlabeled PET probes studied in this paper. (**B**) Secondary structure of tau filaments (K254 to F378) from Pick’s patients showing surface binding sites S1 to S11 (green), cavity site C1 (blue) and neighboring amino acid residues. Each β strand is labeled from β1 to β9.

**Figure 2 ijms-22-00349-f002:**
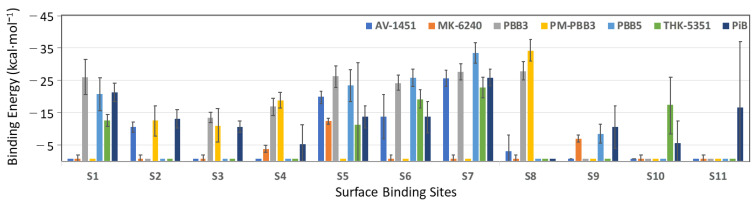
Graph of MM/GBSA binding energies of PET probes in surface binding site (S1 to S11).

**Figure 3 ijms-22-00349-f003:**
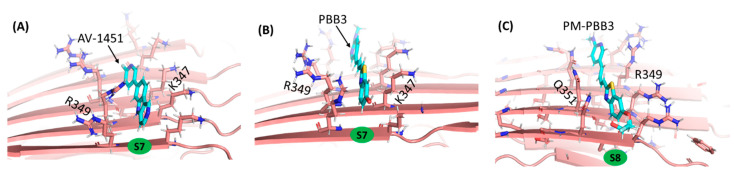
Molecular interactions of PET probes to Tau^PiD^ taken after 100-ns MD simulation. (**A**) AV-1451 and (**B**) PBB3 binding to R349 and K347 sidechain hydrogens in site S7 and (**C**) PM-PBB3 binding to R349 and Q351 sidechain hydrogens in site S8.

**Figure 4 ijms-22-00349-f004:**
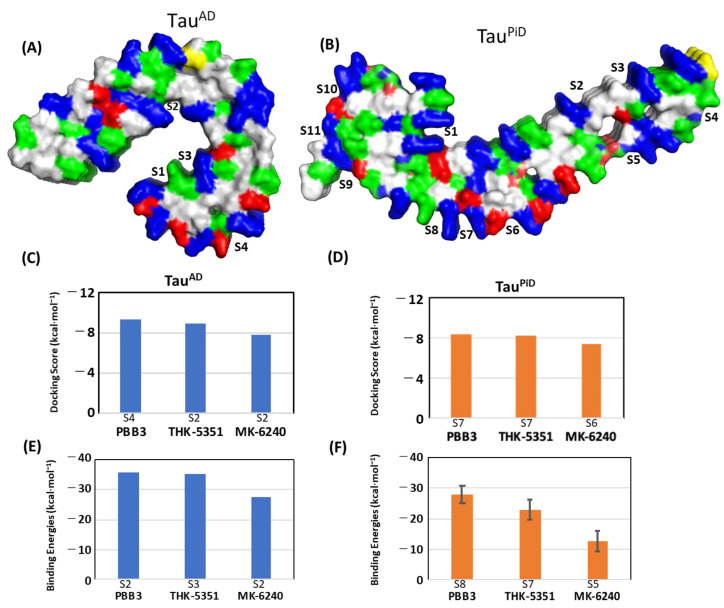
Comparison of the surface binding sites in Tau^AD^ reported previously (**A**) and Tau^PiD^ studied in this paper (**B**). Surface representation for tau structures is colored as residue type (acidic-red, basic-blue, nonpolar-grey, polar-green, cysteine-yellow). Docking scores of the strongest docking pose for PBB3, THK-5351 and MK-6240 in the best scoring site in Tau^AD^ reported in Murugan et al. [25] (**C**) and Tau^PiD^ (**D**). Similarity of MM/GBSA binding energy (kcal·mol^−1^) of the strongest binding modes of PET probes in Tau^AD^ (**E**) and Tau^PiD^ (**F**).

**Figure 5 ijms-22-00349-f005:**
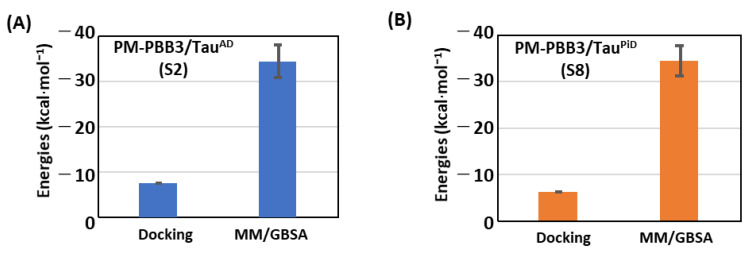
Docking scores and binding affinities of PM-PBB3 in Tau^AD^ (**A**) and Tau^PiD^ (**B**). Standard deviation in MM/GBSA binding energy over all snapshots is shown as error bars.

**Figure 6 ijms-22-00349-f006:**
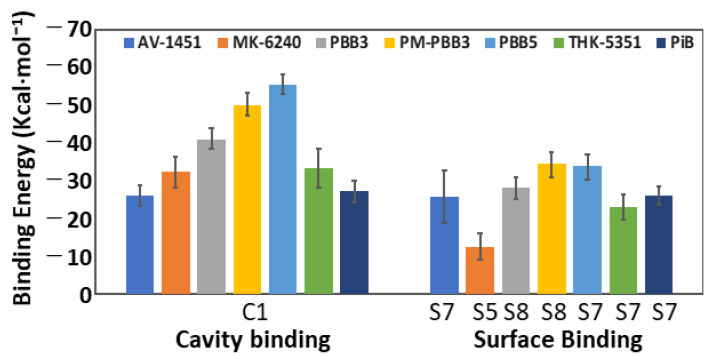
MM/GBSA binding energies of PET probes in site C1 and surface binding sites that show the most potent binding (S1–S11).

**Figure 7 ijms-22-00349-f007:**
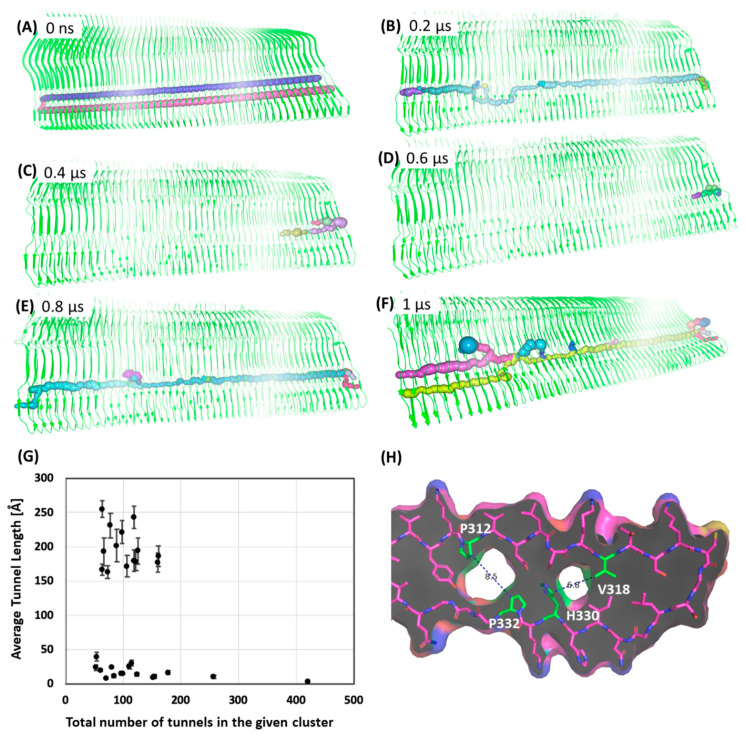
Tunnels predicted by the CAVER program in (**A**) cryo-EM structure and (**B**–**F**) structure taken from MD simulation. The snapshots taken after (**B**) 200 ns, (**C**) 400 ns, (**D**) 600 ns, (**E**) 800 ns, (**F**) 1 microsecond simulation showing tunnels around the cavity site. (**G**) Plot of the tunnel length and number of tunnels found by CAVER on 500 frames (extracted each at 2 ns from 1 microsecond MD simulation) of the unbound tau filament. (**H**) Surface of the Tau^PiD^ structure showing cavities in the structure and distance between neighboring Pro residues.

**Table 1 ijms-22-00349-t001:** Calculated docking scores (kcal·mol^−1^) from molecular docking of various PET probes in the different binding sites of Tau^PiD^ protofibril. Values in brackets are the binding energies from the rescoring function X-Score for the same binding pose. Site C1 corresponds to a cavity site while the remainder (S1 to S11) correspond to surface binding sites. Corresponding binding sites are shown in Figure 1B.

Binding Site	AV-1451	MK-6240	PBB3	PM-PBB3	THK5351	PiB
C1	−11.0 (−8.9)	−11.0 (−9.0)	−10.1 (−9.4)	−10.0 (−9.4)	−10.3 (−9.0)	−8.9 (−8.5)
S1	NB	NB	−4.9 (−7.0)	NB	NB	−4.4 (−6.6)
S2	−6.0 (−7.3)	NB	NB	−5.0 (−7.2)	−5.7 (−7.0)	−4.4 (−6.9)
S3	NB	NB	−5.7 (−7.6)	−5.4 (−7.4)	NB	−4.6 (−7.2)
S4	NB	−5.1 (−6.7)	−5.0 (−6.9)	NB	NB	−4.5 (−6.5)
S5	−6.5 (−7.4)	−5.5 (−6.7)	−6.0 (−7.7)	NB	−5.8 (−7.1)	−5.2 (−7.2)
S6	−8.3 (−7.9)	**−7.4 (−7.7)**	−7.3 (−8.0)	NB	−7.3 (−7.8)	−6.3 (−7.5)
S7	**−9.4 (−8.4)**	NB	**−8.4 (−8.7)**	NB	**−8.3 (−8.3)**	**−7.1 (−7.8)**
S8	−7.1 (−7.5)	NB	−6.2 (−7.6)	**−6.1 (−7.5)**	NB	NB
S9	−5.9 (−6.7)	−5.7 (−6.9)	NB	−4.5 (−6.5)	−5.6 (−6.7)	−4.6 (−6.6)
S10	−6.0 (−7.1)	NB	NB	−4.6 (−6.6)	−5.2 (−6.8)	−4.6 (−6.7)
S11	NB	NB	NB	−4.5 (−6.9)	NB	−4.5 (−6.9)

PM-PBB3 also binds to a site in the left neighbor of site S2 (S2n) with a docking score of −4.6 (−6.7) and a site in the right neighbor of site S3 (S3n) with a docking score of −3.7 (−6.8). NB; did not bind. Lowest scores among all the surface binding sites are in bold.

**Table 2 ijms-22-00349-t002:** The calculated MM/GBSA binding energies (kcal·mol^−1^) of PET probes in Tau^PiD^ binding sites. The label “x” represents sites where PET probe was not docked and a label “-” shows cases where MM/GBSA binding energies could not be calculated to due to tracer drift or steric clashes in certain snapshots. All the energy values are reported in kcal·mol^−1^.

Binding Site	AV-1451	MK-6240	PBB3	PM-PBB3	THK-5351	PiB
C1	−25.8 ± 2.8	−32.1 ± 4.1	−40.9 ± 2.7	−50.0 ± 3.1	−33.2 ± 5.1	−27.0 ± 3.0
S1	x	x	−26.0 ± 5.5	-	x	−21.4 ± 2.9
S2	−10.7 ± 1.6	x	x	−12.7 ± 4.6	−12.6 ± 1.8	−13.1 ± 2.8
S3	x	x	−13.6 ± 1.6	−11.2 ± 5.1	x	−10.7 ± 1.7
S4	x	−3.9 ± 3.7	−16.9 ± 2.6	−18.9 ± 2.4	x	−5.5 ± 5.7
S5	−19.9 ± 1.9	**−12.5 ± 3.3**	−26.2 ± 3.3	x	−11.3 ± 19.3	−13.8 ± 3.4
S6	−13.8 ± 6.9	-	−24.3 ± 2.4	x	−19.3 ± 2.9	−13.7 ± 4.8
S7	**−25.5 ± 2.5**	x	−27.6 ± 2.6	x	**−22.8 ± 3.2**	**−25.9 ± 2.6**
S8	−3.2 ± 5.0	x	**−27.9 ± 2.9**	**−34.2 ± 3.3**	x	x
S9	-	−7.1 ± 5.7	x		-	−10.7 ± 6.5
S10	-	x	x	-	−17.4 ± 8.7	−5.6 ± 6.9
S11	x	x	x	-	x	−16.8 ± 25.8

MM/GBSA binding energy of PM-PBB3 in S2n site (binding site in the left neighbor of S2) (S2n) is −30.2 ± 3.2 kcal·mol^−1^. PM-PBB3 was docked to a site S3n (a site in the right neighbor of site S3 but MM/GBSA energies could not be calculated due to steric clashes. Lowest scores among all the surface binding sites are in bold.

## Data Availability

The data presented in this study are available upon request from the corresponding authors.

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
