# Peer review of "Pick’s Tau Fibril Shows Multiple Distinct PET Probe Binding Sites: Insights from Computational Modelling"

_ijms, 2020, doi:10.3390/ijms22010349_

Round 1

Reviewer 1 Report

The authors report their docking and MD simulations on Tau fibrils. The topic is relevant, as Tau is involved in several neurodegeneratives diseases.The study focus on the binding site and affinity of some PET tracers, which is important to the development of novel tracers for diagnosis.

Concerning the Methods, I believe that should be more clear to the reader that the structures 6GX5 and 5O3L, only comprises 94 and 73 amino acids, respectively, from the total 441 residues present in 2N4R Tau sequence.

These Cryo-EM structure correspond to which part of Tau? (N-terminal, Proline-rich, microtubule binding region, C-terminal) And why its is relevant in the formation of fibrils and for the interaction with tracers.

Also, it is not clear for me how the docking complexes evolved to MD simulations. That is, the docking positions were transferred to the simulation, one by one, or, together, one tracer in all docked positions was associated (designed) at once with the Tau excerpt?

The Methods (protocol) for docking, MD simulations, Caver, and MM/GBSA are solid and conducted properly.

The resolution of Figure 1(A) and Figure 7 can be improved.

Typo in line 55: Serval > Several...

I support the publication of this manuscript with minor revisons.

Author Response

Response to Reviewer 1 Comments

Point 1: Concerning the Methods, I believe that should be more clear to the reader that the structures 6GX5 and 5O3L, only comprises 94 and 73 amino acids, respectively, from the total 441 residues present in 2N4R Tau sequence.

Response 1: Thank you for the comment, we added a sentence at line 474, “TauPiD and TauAD structures comprises 94 and 73 amino acids, respectively, from the total 441 residues in 2N4R tau sequence.”

Point 2: These Cryo-EM structure correspond to which part of Tau? (N-terminal, Proline-rich, microtubule binding region, C-terminal) And why its is relevant in the formation of fibrils and for the interaction with tracers.

Response 2: The cryo-EM structures revealed the C-terminal core region of Tau. We modified a sentence at line 42, “Electron cryo-microscopy (cryo-EM) has revealed high-resolution structures of C-terminal core region in tau filaments from AD, PiD, CBD and chronic traumatic encephalopathy (CTE) brains”

Point 3: Also, it is not clear for me how the docking complexes evolved to MD simulations. That is, the docking positions were transferred to the simulation, one by one, or, together, one tracer in all docked positions was associated (designed) at once with the Tau excerpt?

Response 3: We added a sentence at line 495, “For each PET probe, we performed MD simulation of fully solvated assemblies where PET probe is simultaneously bound to TauPiD in all the binding sites predicted by docking.”

Point 4: The resolution of Figure 1(A) and Figure 7 can be improved.

Response 4: We have replaced Figure 1(A) and Figure 7 with new figures having improved resolution.

Point 5: Typo in line 55: Serval > Several...

Response 5: Thank you. We have corrected this typo.

Reviewer 2 Report

The authors present in silico approaches to define the binding of PET tracers to Tau fibrils from rare age-related dementias like Pick disease. There is plethora of work on understanding the interactions between tracers and tau filaments from AD but there is definite knowledge gap when it comes to understanding tau tracer binding in more uncommon tauopathies. Therefore, this computational proof of concept work is a step in the right direction to understand the tau tracer binding to tau fibrils. 

However, this reviewer has some concerns about the off-target binding. Some of the first generation tau tracers used in this study are known to show non-specific off-target binding. It would be a good comparison to show the binding characteristics of the tracers used in this study to a non-tauopathy target like MAO-A or even other dementia/neurodegenerative proteins like amyloid-β or α-synuclein using the same tools used in this study. The authors compare the binding of AD-tau and PiD-tau, which is a good comparison to have, but adding a non-specific target would enhance the significance of the processes outlined here. 

Other than this minor suggestion, this reviewer thinks that the manuscript is well written and can be accepted for publication once the authors address the concern.

Author Response

Point 1: This reviewer has some concerns about the off-target binding. Some of the first generation tau tracers used in this study are known to show non-specific off-target binding. It would be a good comparison to show the binding characteristics of the tracers used in this study to a non-tauopathy target like MAO-A or even other dementia/neurodegenerative proteins like amyloid-β or α-synuclein using the same tools used in this study. The authors compare the binding of AD-tau and PiD-tau, which is a good comparison to have, but adding a non-specific target would enhance the significance of the processes outlined here. 

Response 1: Many thanks for pointing this out. We agree that studying binding mechanisms of PETs on systems would be interesting to explore the off-target effects. However, our study was designed to focus specifically on Tau filaments this time and we'd like to keep the focus intact. We have discussed off target effects of the PETs from literature and included this point as a consideration for future study (lines 331-336).